# Epidemiology of Chronic Hepatitis B Infection in the Cohort of College Students with Vaccination in Taiwan

**DOI:** 10.3390/vaccines11020348

**Published:** 2023-02-03

**Authors:** Te-Wei Cheng, Jeng-Fu Yang, Yi-Yu Chen, Kuan-Ta Wu, Meng-Szu Lee, Hsiang-Ju Kuo, Tzu-Chun Lin, Chao-Ling Wang, Meng-Hsuan Hsieh, Chia-Yi Lin, Batbold Batsaikhan, Chi-Kung Ho, Chia-Yen Dai

**Affiliations:** 1School of Medicine, College of Medicine, Kaohsiung Medical University, Kaohsiung 80756, Taiwan; 2Health Management Center, Department of Occupational and Environmental Medicine, Kaohsiung Medical University Hospital, Kaohsiung Medical University, Kaohsiung 80756, Taiwan; 3Department of Public Health, College of Health Sciences, Kaohsiung Medical University, Kaohsiung 80756, Taiwan; 4Executive Master of Healthcare Administration, Department of Healthcare Administration and Medical Informatics, Kaohsiung Medical University, Kaohsiung 80756, Taiwan; 5Hepatobiliary Division, Department of Internal Medicine, Kaohsiung Medical University Hospital, Kaohsiung Medical University, Kaohsiung 80756, Taiwan; 6Department of Internal Medicine, Institute of Medical Sciences, Mongolian National University of Medical Sciences, Ulaanbaatar 14210, Mongolia; 7Drug Development and Value Creation Research Center, Kaohsiung Medical University, Kaohsiung 80756, Taiwan

**Keywords:** HBV, vaccine, HBsAg, anti-HBs

## Abstract

After the mass vaccination project in Taiwan, the prevalence of the hepatitis B virus (HBV) infection for the college-aged population of 18 to 21 years is uncertain. We aimed to investigate the prevalence of hepatitis B markers in different birth cohorts. A total of 38,075 students in universities in Kaohsiung area undergoing entrance examinations between July 2006 to September 2020 were included. Seroprevalence of the hepatitis B surface antigen (HBsAg) and hepatitis B surface antibody (anti-HBs) status and laboratory data were collected. The seropositive rate of HBsAg was less than 1% for students born after 1991. Alanine aminotransferase (ALT) and aspartate aminotransferase (AST), were significantly higher, and body mass index (BMI) was significantly lower in HBV carriers compared to those who were not carriers (all *p* < 0.001). Multivariate logistic regression showed that age, male, higher BMI, and positive HBsAg were risk factors of abnormal ALT value. A decrease in the positive rate of anti-HBs which was significantly higher in the cohort of plasma-derived vaccines than recombinant vaccines was found. We concluded that there were decreasing trends in seropositive rates of HBsAg and anti-HBs for students of the college-aged population in the Kaohsiung area. The status of HBsAg was a predictive factor of abnormal ALT levels. The period effect on anti-HBs seropositivity for DNA recombinant vaccine somehow existed.

## 1. Introduction

Hepatitis B virus (HBV) infection is a global health issue. According to the surveys of the World Health Organization (WHO) in 2021, the overall incidence of new infections of HBV was 1.5 million per year, and it was estimated that for the population of children under 5 years old, the prevalence was less than 1% [1]. Aside from the risk of fulminant hepatitis, HBV may lead to liver cirrhosis and the development of hepatocellular carcinoma. Moreover, it would facilitate the coinfection of Hepatitis D virus (HDV) [2,3,4].

Since the launch of the national vaccination program in 1984, the seroprevalence rate of HBV declined from 9.8% to less than 1% in Taiwan since 2009 [5,6,7]. Meanwhile, some studies also revealed that there were trends of decreasing level of the hepatitis B surface antibody (anti-HBs) for people who had received the HBV vaccines with age [8]. Additionally, there also seemed to be a declining trend in the seropositivity of the anti-HBs as people were born in later birth years; that is, anti-HBs seropositivity of the population in the postvaccination cohort at around 2–4 years were higher than that of the population in the postvaccination cohort at 6–8 years. It was also found much lower at 15–18 years [6,9]. Tracing back the history of the universal HBV vaccination program, it was not until July 1984 that the newborns of high-risk mothers whose HBsAg was positive received vaccination. Before November 1992, plasma-derived vaccines were adopted. After then, the recombinant vaccines were used [10]. Although there was a waning in titer of anti-HBs, immunity, for the vaccinated people with age, it appeared that there was no increase in HBV infection. Hence, booster vaccination was not a mandatory program for newborns according to the Center for Diseases Control (CDC). It was not a universal booster but an individual booster that needed adopting. According to recommendations from the WHO, there was no evidence supporting the need for the immunocompetent to receive a booster after primary vaccination [11]. There were numerous studies elucidating what dose is required to reactivate sufficient anti-HBs for protectivity [12,13]. Nevertheless, few pieces of research focused on how the trends of waning in anti-HBs titer changed. Furthermore, it was reported that more than half of the vaccinees had undetectable anti-HBs after 15 to 18 years of vaccination. In a study in Taiwan, a cohort group of high school students born after 1986 seemed to have lower frequency of positive HBsAg, hepatitis B core antibody (anti-HBc), and anti-HBs [14,15]. This implied that freshmen in college without revaccination by booster could be susceptible to HBV infection. There has been no recent study specifically aimed at the school-age population of college and grouped by birth cohort. On the other hand, though surveillance reports for HBV vaccination program in Taiwan for every 5 years are available, these surveys mainly focused on the Taipei metropolitan area. It seems that there has been no comprehensive survey in southern Taiwan.

The present retrospective study aimed to figure out how the universal HBV vaccination program impacted the school-age population of the college-aged 18 to 21 year-olds according to different birth cohorts in the Kaohsiung area, the largest city in southern Taiwan.

## 2. Materials and Methods

### 2.1. Patients and Materials

We collected records of entrance examinations for freshmen who enrolled in three universities in the Kaohsiung area from July 2006 to September 2020. All undergraduates and graduate students would undergo one-time entrance examinations during the enrollment in the first semester. The items of student health examinations included body weight, body height, survey of general systems, laboratory tests with blood tests, and HBsAg status and anti-HBs status were included for screening of HBV infection. Data were archived in the database center of Kaohsiung Medical University. In this study, all data were handled by de-identification to break the link between data and the individual before the analysis. Moreover, duplicate data for students with the same identification number for reasons such as repeated enrollment in different year would be eliminated. Since we focused on the period of college age, age range of included subjects was from 18 to 21 years old, which met the criteria of college students defined by National Development Council. A total of 38,075 students were eventually included in this retrospective study with different enrolling years. The study was approved by the institutional review board (IRB) of Kaohsiung Medical University Hospital (KMUHIRB-E(I)-20210290).

The titers of serum anti-HBs and HBsAg were tested by using standard quantitative chemiluminescent microparticle immunoassay (ARCHITECT HBsAg, Abbott Diagnostics) or qualitative assay (Abbott Laboratories, North Chicago, IL, USA). For anti-HBs, titer < 10 IU/L (mIU/mL) was considered seronegativity; on the other hand, titer ≥ 10 IU/L was considered seropositivity. HBV immunity or protection against hepatitis B virus infection would be defined as students with seropositive anti-HBs and seronegative HBsAg. HBV carrier was defined as students with positive status of HBsAg.

### 2.2. Statistical Analysis

R version 4.0.4 was adopted for the statistical analysis. Descriptive statistics included age, gender, anti-HBs status, HBsAg status, alanine aminotransferase (ALT), aspartate aminotransferase (AST), total cholesterol, body height, body weight, and body mass index (BMI). Frequencies were compared using the χ^2^ test with the Yates correction or Fisher’s exact test. Group means were presented as mean ± standard deviation or proportion as percentage (%) and were compared by using Student’s *t*-test, analysis of variance, or the nonparametric Mann–Whitney test. We compared groups between HBV infection and non-HBV infection based on HBsAg status in the first place. Univariate analyses were performed first in variables studied in descriptive statistics (age, gender, ALT, AST, total cholesterol, body height, body weight, and BMI) to identify the factors related to HBsAg status. A multivariate analysis was then used with the same variables as covariates in continuous form. BMI was the function of body height and body weight, and it would be the variable included in model of multivariate analysis. We also introduced the concept of clinical criteria for obesity with threshold of BMI not less than (≥) 24 kg/m^2^ according to Health Promotion Administration, Ministry of Health and Welfare and normal liver enzyme with ASSLD 2018 Hepatitis B Guidance of ALT cutoffs: male: ≤35 U/L; female: ≤25 U/L into analysis. Multivariate logistic regression for abnormal ALT status was also adopted to find the impact of well-accepted predictive factors: age, gender, and BMI, plus HBsAg status. Two outcomes variables with laboratory abnormal ALT cut-off level (>44 U/L) and abnormal ALT of ASSLD cut-off level were conducted separately. The odds ratio was presented with 95% confidence interval (CI). In order to figure out the impact on birth cohort effect, subjects were classified into three groups according to the birth year and the type of vaccines: Group I: between 1985 and October 1992, the plasma-derived vaccines were adopted; Group II: between November 1992 and 1996, the recombinant vaccines were used; Group III: between 1997 and 2002, the recombinant vaccines were also used. Two-tailed *p*-value < 0.05 was considered statistically significant.

## 3. Results

### 3.1. Characteristics of Subjects

Table 1 shows details of the seroprevalence of HBV infection and immunity of the 38,075 students aged 18 to 21 years according to birth year. All subjects were post-vaccination cohort with birth year later than 1984. Overall, the seropositive rate of HBsAg was maintained at less than 1% for students born after 1991. Moreover, there was no chronic HBV carrier for the youngest two birth years: 2001 and 2002. The trend of decline in seropositivity of HBsAg was depicted in Figure 1. It reached 7.84% overall in the students born in 1985 and decreased progressively to 0 for population born after 2001. Among students whose birth years were 1985, 1992, and 2002, the HBV immunity was 74.5%, 49.2%, and 38.2%, respectively. Overall, HBV immunity decreased with years, but short-term rebound was found in 1990, 1991, 2000, and 2001 (Figure 2).

When comparing trends among genders, for male students, the seropositivity of HBsAg was 11.54% for those who were born in 1985, and it decreased to less than 1% after 1991. The seropositivity of HBsAg in the female population decreased from 4% in the birth year of 1985 to less than 1% since the birth year 1991 (Figure 1). From the birth year 1985 to 2002, the trend of decreasing HBV immunity from 73.1% to 38.6% was noted in males. The same tendency for the decline was also noted in females from 76.0% to 37.9% from the birth year of 1985 to 2002. In addition, we also found transient increasing trends in 1987, 1990, 1991, 2000, and 2001 in the male group. In the female group, this tendency was noted in 1989, 1990, 1991, and 2000 (Figure 2).

### 3.2. Increase in ALT, AST, and BMI for HBV Carrier Subjects

When analyzing depending on HBsAg status, in univariable analysis, there was no difference in proportion of gender, age, height, weight, BMI, proportion of obesity, and total blood cholesterol between the group of HBV carriers and the group of non-HBV infection. On the other hand, the serum ALT and AST were significantly different in univariable analysis. These two liver enzymes were higher in the population of HBV carriers. Furthermore, proportion of abnormal ALT based on AASLD criteria was also significantly different. It seemed that higher proportion of HBV carriers were with abnormal status of ALT with clinical significance. Multivariate analysis for continuous variables revealed ALT and AST were significantly higher, and the BMI was significantly lower in the population of HBV carriers. (Table 2).

### 3.3. Impacts of Predictive Factors on Abnormal ALT Status with ASSLD Criteria

ALT was known as the liver enzyme more specific to liver damage. Multivariate logistic regression for abnormal ALT status based on the laboratory and ASSLD criteria was adopted with predictive factors of age, gender, HBsAg status, and BMI. It was compared with results of multivariate logistic regression for abnormal ALT status based on general cutoff (Table 3).

When it came to gender, males had an adjusted odds ratio (ORadj) of 1.55 (95% CI, 1.43–1.68). There was an around 1.6-fold greater odds of abnormal ALT status for males than for females. Compared to age, an increase in age by each year was about 1.1-fold greater odds of abnormal ALT status (ORadj = 1.06, 95% CI, 1.01–1.11). Body mass index was another predictive factor with adjusted odds ratio of 1.30 (95% CI, 1.29–1.31), which meant that with an increase in body mass index by each unit, there was a 1.3-fold greater odds of abnormal ALT status. HBsAg was the strongest predictive factor of ALT status. The adjusted odds ratio was 5.85 (95% CI, 4.19–8.16). Chronic HBV carriers were about six times more likely to be abnormal in ALT status (Table 3). These results of ALT by ASSLD criteria were in accordance with that of laboratory cutoff of abnormal ALT.

### 3.4. Cohort Effect in College Students

At college age (18–21 years), the seroprevalence of anti-HBs in non-HBV carriers significantly decreased from 49.2% for cohort group 1 to 42.1% and 35.8% for cohort group 2 and 3, respectively (Figure 3) (chi-square test, *p* < 0.001; post-hoc: cohort 3 > cohort 2 > cohort 1, *p* < 0.001).

## 4. Discussion

In this retrospective study, like other studies, there were similar trends of decreasing seroprevalence of hepatitis B surface antigen after the national hepatitis B vaccination program in 1984 in Taiwan [7,10]. Because we focused on the population born after July 1984, the seropositive rates of HBsAg and anti-HBs more precisely showed the trends in the vaccinated cohorts. Especially, in the latest two birth years, 2001 and 2002, there were zero chronic HBV carriers when students entered college. Age–period–cohort analysis was used for incidence of HBV infection in previous studies. There were many modified models with trivially different assumption for this analysis, but it was comprised of three core components: age effects, period effects, and cohort effects. Age effects, compared to period and cohort effects, which reflected the nature of aging, seemed to be the most deterministic factor when it came to chronic HBV infection in human, while period effects reflected influences on all age groups at a particular timing, and cohort effects, commonly defined as birth cohort, could reflect the implementation of the vaccination in history of HBV infection. [6,16]. Hence, in our studies, we focused on the school-aged population of the college, from 18 to 21 years, which was defined by our National Development Council. In this way, we could observe differences among cohorts more specifically.

Hepatitis B virus infection was one of the common causes of elevated transaminase levels. There were four phases of chronic hepatitis B infection due to its nature of dynamics: immune-tolerant phase, hepatitis B e antigen (HBeAg)-positive immune-active phase, inactive CHB phase, and then HBeAg-negative immune reactivation phase. We could clarify which phase the subject was in depending on ALT level, HBV DNA level, status of HBeAg, and liver histology profile [17]. In our study, we found ALT level was increased in HBV infection group. Elevated ALT often accounted for HBeAg-positive immune-active phase and HBeAg-negative immune reactivation phase. It could be related to increased immune reaction with damage of the hepatocytes and led to increased liver fibrosis in this population. 

ALT, the marker of hepatocellular injury, seemed to be higher in the population of HBV carriers. Moreover, carriers were more likely to be abnormal in clinical criteria. BMI, age, and sex were also found to be relative to the abnormality of ALT, but HBsAg status was the strongest [18,19]. Therefore, it was urgent for those college students with positive HBsAg to accept a series of further surveys, such as tests for HBV replication, liver chemistry tests, and surveillance of liver cirrhosis. With comprehensive examinations, college students may need to receive suitable medical care such as oral antiviral drug depending on the level of ALT, DNA viral load, and presence of liver cirrhosis, or the disease could exacerbate according to the guidelines [18,20]. Although it was not surprising that increased level of ALT is related to age, male, obesity, and HBsAg status, in our studies, we tried to introduce ASSLD criteria to the college-aged population. Hence, we could clarify how much contribution these risk factors made.

It was noteworthy that the seroprevalence of anti-HBs decreased from the early cohort to the late cohort. Previously, many studies for the population of college students with primary vaccination during infancy were done worldwide. The ratio of protective anti-HBs level ranged from 55.0% (mean age: 20.7 years old) in Italy, 49% (mean age: 22.8 years old) in Saudi Arabia, to 41.3% (mean age: 21.7 years old) in Israel [21,22,23]. In Romania, there was a study for anti-HBs antibody titers in vaccinated birth cohorts. A universal newborn vaccination program was initiated since 1995 in Romania. For birth cohort with primary vaccination at birth, seroprevalence of anti-HBs was 41.3% in the cohort with birth year between 1995 to 2006 (mean age: 20 years old and mean time since vaccination: 17.73 years). In contrast, for the later cohort with birth year since 2007 (mean age: 5 years old and mean time since vaccination: 5.75 years), seroprevalence of anti-HBs was 67.0%. However, the age effects could not be neglected [24]. It seemed that there were no studies focusing on different birth cohorts with the population of college students. 

As in other studies in Taiwan, the population with plasma-derived vaccine cohort maintained a higher proportion of immunity, in contrast to the population with recombinant vaccine [25,26,27,28]. However, we also found that even after receiving recombinant vaccines, there was still a trend of decrease in anti-HBs rate. There seemed to be other factors that led to this waning immunity. Although studies in other countries reported that body mass index or vitamin D status might be related to the persistence of anti-HBs antibody, there was no such trend about BMI and anti-HBs antibody in our survey. Moreover, there were no data on Vitamin D in the database to support it. However, owing to the fact that these populations belonged to the same college age group, 18 to 21 years, and accepted recombinant vaccines, it could be the period effect that had some influence on anti-HBs antibody seropositivity [29,30]. 

Two brands of the recombinant HBV vaccines had been adopted since November 1992: Engerix-B^®^ (GlaxoSmithKline, GSK) and HB-VAX II^®^ (Merck Sharp and Dohme, MSD), and both were made of the same compartment of protein—small HBsAg protein comprising a 226 amino acid, called S-domain [31]. Up to now, every newborn is given 3 sequential doses of recombinant vaccines at the age of 24 h, 1 month, and 6 months after birth. Some experts are concerned about the efficacy of HBV vaccines in the long run due to the fact that the prevalence of the mutation of a determinant (121st–149th amino acid) of HBsAg increased since the post-vaccination era [32,33,34]. Hence, further surveys are needed to clarify if there is a relationship between the prevalence of the mutation of a determinant of HBsAg and the seropositivity of anti-HBs. Further studies are mandatory from both the immunology and the viralogical aspects. 

Effects of delay vaccination of 1st and 3rd dose were correlated with risk of having a titer of anti-HBs below 10 IU/L after 20 years of primary vaccination during infancy in a study in Italy. A −16% and −11% in immunity was reported for one month delay of 1st and 3rd dose. This effect of delay was not noted in the group of primary vaccination during adolescence with mean age of vaccination around 11 years old [35]. In our studies, interval between doses was assumed to be 24 h, 1 month, and 6 months after birth. Furthermore, the differences in the vaccination rate between rural and urban areas are generally considered small. Hence, we believe our study results from Kaohsiung can be generalized to the newly enrolled college students in Taiwan. The study results are also considered consistent with the HBV surveillance reports in Taiwan: A decreasing trend of HBsAg seropositivity rate and a decreasing tendency of anti-HBs seropositivity rate in the postvaccination cohort. The HBV surveillance policy for college students in Taiwan may need further study in the future, even though the prevalence of the HBsAg is very low with the disappearance of anti-HBs with age. 

There were some limitations in our studies. First of all, no comprehensive history of intervention of hepatitis B immune globulin (HBIG) and antiviral therapy was collected. Owing to the fact that mother-to-infant transmission was the major pathway of HBV transmission in Taiwan, government-funded HBIG would be merely available for people with positive maternal HBeAg. Whether the prophylaxis of HBIG for vertical transmission worked or not impacted the following HBsAg status of newborns [36,37]. Secondly, though two brands of recombinant vaccines in the national vaccination program were designed with the same logic, the details of the distributions and implantations could still have impacts on the duration of the persistence of anti-HBs. However, it was not accessible in our studies. We will try to clarify them in the future studies. Lastly, there was no hepatitis B core antibody (anti-HBc) in our database. However, since our study population was those born in a post-vaccination cohort, the contribution of a booster effect via natural infection to persistent rate of protectivity could be considered quite low. Nonetheless, we will conduct studies to elucidate this issue in the future.

## 5. Conclusions

In conclusion, in our retrospective study, there were decreasing trends in seropositive rates of HBsAg and anti-HBs for students of the college-aged population in the Kaohsiung area. Additionally, the status of HBsAg was the important predictive factor leading to the abnormal liver enzyme. The period effect on anti-HBs antibody seropositivity for DNA recombinant vaccine somehow existed which might need further studies in the future.

## Figures and Tables

**Figure 1 vaccines-11-00348-f001:**
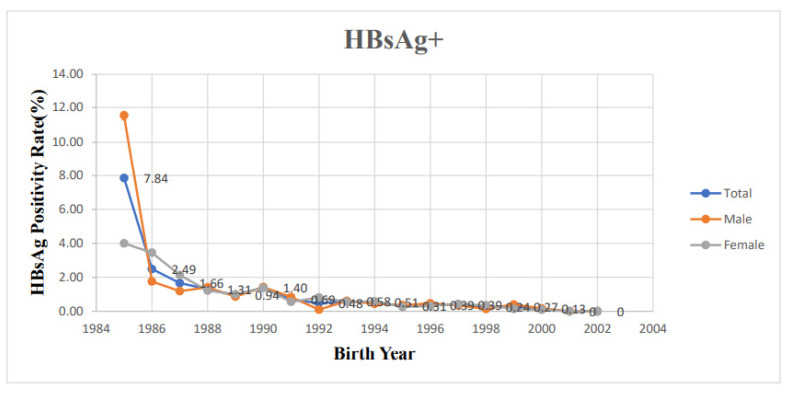
Positive rate of the Hepatitis B surface antigen according to the birth year.

**Figure 2 vaccines-11-00348-f002:**
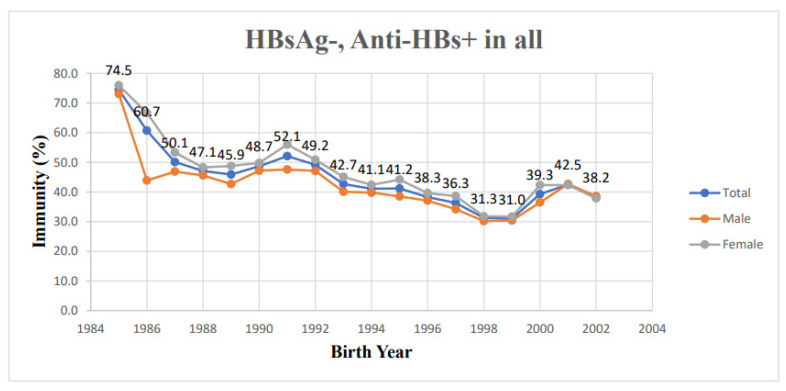
Positive rate of antibody against hepatitis B surface antigen according to the birth year.

**Figure 3 vaccines-11-00348-f003:**
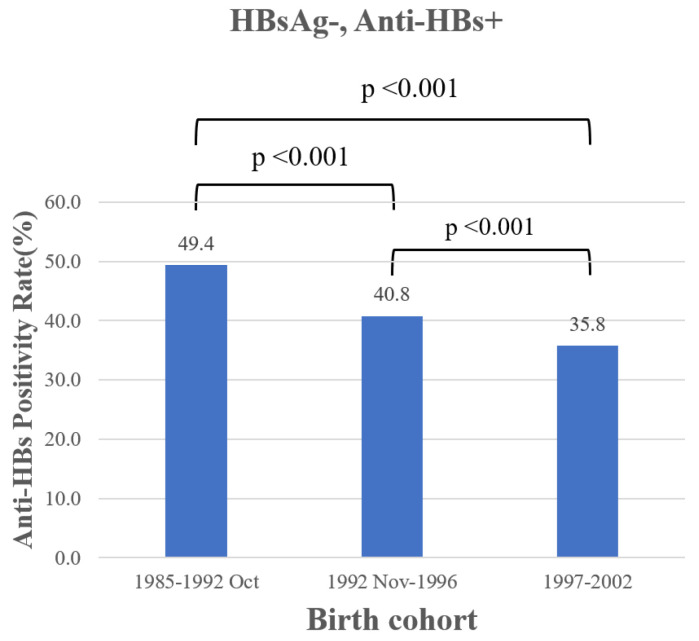
Anti-HBs status in birth cohort.

**Table 1 vaccines-11-00348-t001:** Hepatitis B virus infection according to birth year of study sample.

Birth Year	Sex			HBsAg+			HBsAg-, Anti-HBs+		
	Total	Female	Male	Total	Female	Male	Total	Female	Male
	n	n	n	n (%) ^1^	n (%) ^2^	n (%) ^3^	n (%) ^1^	n (%) ^2^	n (%) ^3^
1985	51	25	26	4 (7.84)	1 (4.00)	3 (11.54)	38 (74.5)	19 (76.0)	19 (73.1)
1986	201	87	114	5 (2.49)	3 (3.45)	2 (1.75)	122 (60.7)	58 (66.7)	64 (43.9)
1987	844	422	422	14 (1.66)	9 (2.13)	5 (1.18)	423 (50.1)	225 (53.3)	198 (46.9)
1988	2144	1149	995	28 (1.31)	14 (1.22)	14 (1.41)	1009 (47.1)	555 (48.3)	454 (45.6)
1989	1708	896	812	16 (0.94)	9 (1.00)	7 (0.86)	784 (45.9)	437 (48.8)	347 (42.7)
1990	1787	1014	773	25 (1.40)	14 (1.38)	11 (1.42)	870 (48.7)	505 (49.8)	365 (47.2)
1991	2330	1253	1077	16 (0.69)	7 (0.56)	9 (0.84)	1215 (52.1)	702 (56.0)	513 (47.6)
1992	2276	1230	1046	11 (0.48)	10 (0.81)	1 (0.10)	1119 (49.2)	626 (50.9)	493 (47.1)
1993	2408	1257	1151	14 (0.58)	7 (0.56)	7 (0.61)	1028 (42.7)	567 (45.1)	461 (40.1)
1994	2731	1368	1363	14 (0.51)	8 (0.58)	6 (0.44)	1123 (41.1)	580 (42.4)	543 (39.8)
1995	3230	1558	1672	10 (0.31)	4 (0.26)	6 (0.36)	1332 (41.2)	688 (44.2)	644 (38.5)
1996	3370	1640	1730	13 (0.39)	5 (0.30)	8 (0.46)	1292 (38.3)	650 (39.6)	642 (37.1)
1997	3369	1636	1733	13 (0.39)	7 (0.43)	6 (0.35)	1224 (36.3)	632 (38.6)	592 (34.2)
1998	2897	1436	1461	7 (0.24)	5 (0.35)	2 (0.14)	908 (31.3)	467 (31.8)	441 (30.2)
1999	2958	1449	1509	8 (0.27)	2 (0.14)	6 (0.40)	917 (31.0)	459 (31.7)	458 (30.4)
2000	2290	1086	1204	3 (0.13)	1 (0.09)	2 (0.17)	899 (39.3)	460 (42.4)	439 (36.5)
2001	1963	1013	950	0 (0)	0 (0)	0 (0)	835 (42.5)	429 (42.3)	406 (42.7)
2002	1518	808	710	0 (0)	0 (0)	0 (0)	580 (38.2)	306 (37.9)	274 (38.6)

^1^ Percentage of total population in the birth year; ^2^ Percentage of total female population in the birth year; ^3^ Percentage of total male population in the birth year.

**Table 2 vaccines-11-00348-t002:** Descriptive analysis of study sample according to HbsAg status.

Variables	HBV Infection		Univariate Analysis	Multivariate Analysis
	No (N = 37,874)	Yes (N = 201)	*p*-Value	*p*-Value
Age, years, mean ± SD	18.38 ± 0.76	18.48 ± 0.87	0.108	0.062
Sex				
Male, N (%)	19,221 (99.5%)	106 (0.5%)	0.623 ^3^	0.259
Female, N (%)	18,653 (99.5%)	95 (0.5%)		
Height, cm, mean ± SD	166.1 ± 8.58	166.0 ± 8.19	0.782	-
Weight, kg, mean ± SD	59.96 ± 12.87	59.16 ± 10.96	0.304	-
BMI, kg/m^2^, mean ± SD	21.61 ± 3.71	21.42 ± 3.37	0.414	0.005
<24, N (%)	30,085 (99.5%)	163 (0.5%)	0.622 ^3^	
≥24, N (%)	7789 (99.5%)	38 (0.5%)		
ALT ^1^, U/L, mean ± SD	18.24 ± 20.00	35.17 ± 66.09	<0.001	<0.001
ASSLD ^2^				
Normal, N (%)	34,347 (99.6%)	142 (0.4%)	<0.001 ^3^	
Abnormal, N (%)	3527 (98.4%)	59 (1.6%)		
AST ^1^, U/L, mean ± SD	20.83 ± 10.23	27.94 ± 24.70	<0.001	0.045
Total cholesterol, mg/dL, mean ± SD	173.5 ± 30.73	175.3 ± 29.61	0.398	0.979

^1^ ALT, AST: Disobeying normality (Using log-transformation); ^2^ AASLD 2018 Hepatitis B Guidance, criteria of ALT cutoffs: male: ≤35 U/L; female: ≤25 U/L; ^3^ Chi-square test.

**Table 3 vaccines-11-00348-t003:** Descriptive analysis and multivariate analysis logistic regression for factors associated with laboratory abnormal ALT status (>44 U/L), and abnormal ALT of ASSLD criteria (female > 25; male > 35 U/L) in the students.

Model	Abnormal ALT			Abnormal ALT of ASSLD Criteria		
	Mean (SD)	Adjusted Odds Ratio	*p*-Value	Mean (SD)	Adjusted Odds Ratio	*p*-Value
Gender (male/female)	18,748/19,327	3.92 (3.44–4.46)	<0.001	18,748/19,327	1.55 (1.43–1.68)	<0.001
HBsAg	201/37,874	5.56 (3.62–8.54)	<0.001	201/37,874	5.85 (4.19–8.16)	<0.001
Age, year	18.38 (0.73)	1.09 (1.02–1.16)	0.012	18.38 (0.73)	1.06 (1.01–1.11)	0.021
BMI, kg/m^2^	21.61 (3.71)	1.30 (1.29–1.32)	<0.001	21.61 (3.71)	1.30 (1.29–1.31)	<0.001

## Data Availability

The data presented in this study are available upon request from the corresponding author.

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
