# Peer review of "Epidemiology of Chronic Hepatitis B Infection in the Cohort of College Students with Vaccination in Taiwan"

_vaccines, 2023, doi:10.3390/vaccines11020348_

Round 1

Reviewer 1 Report

Major notes:

The authors' main finding was that a significant decrease in the rate of anti-HBs positivity was found by birth cohort. The period effect on anti-HBs antibody seropositivity for DNA recombinant vaccine somehow existed which might need further studies in the future.

The disadvantage of the paper is that this finding is not particularly justified. The authors report similar findings in other papers (citations 17-20), but all of them were done in sets of students from Taiwan, i.e. on a very similar population. The idea is offered whether the reason for this decrease according to birth cohort is not the fact that, in parallel, HBsAg positivity among students and apparently in the whole society decreased significantly in the corresponding years, and thus the "booster" effect after natural infection gradually decreased. Such a possibility cannot be ruled out even based on the findings of Kao JT et all (cit. 19), who found at the same time a significantly decreasing trend of anti-HBc positivity, which is a sign of natural infection - i.e. probable "boosting" in persons protected by the vaccine. The contribution of the work would be increased if the author added the analysis of anti-HBc antibodies in the same birth cohorts.

Other findings of the authors were not new, unexpected, or unusual, on the contrary, these results were to be expected, they have known reasons and similar data have already been published.

Minor formal notes:

1.    Line 97: „the students born in 1986 and decreased progressively to 0.“ I think the correct year should have been 1985.

2.    In the text, aminotransferases are referred to as ALT and AST, on the contrary in tab. 2 and tab. 3 as sGPT and sGOT. A uniform designation throughout the manuscript would be appropriate.

Reviewer 2 Report

The authors conducted a retrospective cohort study to assess the impact of HBV vaccination program for the cohorts aged 18-21, enrolled in the universities in Kaohsiung, Taiwan. It will be helpful if the authors can better clarify the study design, justify the study population chosen, clarify the analysis method and generalization of the results.

Line 46, is the recent decade referred to 2010s?

Line 50, what years are “later birth years”?

Line 60, since HBV surveillance report in Taiwan is available, the authors mentioned that there was no study specifically aimed at the newly enrolled college students recently, why is this sub-population important for HBV infection control and prevention?

Line 70, what is the rate of participation among the freshmen enrolled in Kaohsiung are from 2006 to 2020?

Line 74, please clarify how the data were collected. For example, how were the tests given? How were the test results collected? Were the tests done consistently overtime?

Line 78, the statistical analysis section needs to be further clarified and should be aligned with the results. Many analyses presented in the results section were not stated in this section. For different analysis done, the authors also need to better clarify what the comparison groups were.

Line 97, please clarify when seropositivity was declined to 0.

Line 117-Line 119, the univariate and multivariable analysis should be included in the method section.  It is also unclear why ASSLD is not included in the multivariable analysis since the factor is statistically significant in the Univariate analysis. The author might also want to keep the terminology (i.e. ASL/GOT and ALT/GPT) consistent in the text and table consistent.

Line 129-131, the authors stated that “there were significant differences 129 in age, height, weight, BMI, total cholesterol and HBV status in univariate analysis”, were additional analysis done to obtain the results? Was this analysis done based on participants who were positive? The analysis conducted need to be included in the method. Also, the results were not in Table 2.

Line 133-140, the results were not aligned with Table 3 and the interpretation of the results were questionable. Are they odds ratios or relative risk ratios?

Line 157-159, were age effects similar here similar to period or cohort effects? It is not clear what the differences are.

Line 166, As I know, for HBV carriers, it is recommended to have regular check-up and good lifestyle to prevent disease progression, there is no treatment available for HBV carriers to clear the HBV infection. Please clarify the treatment you are referring. What are the criteria?

As the authors stated in the title that this is a study for college studies in Taiwan, the authors also mentioned that the study participants were newly enrolled college students in Kaohsiung which is the largest city in southern Taiwan, can the study results from Kaohsiung be generalized to the newly enrolled college students in Taiwan and why?  Are the study results consistent with the HBV surveillance report in Taiwan?  What can this study add to the existing HBV surveillance in Taiwan?

Reviewer 3 Report

1. It is a very carefully prepared manuscript, which belongs to the journal Vaccine

2. The authors screened a large cohort of 38,075 students for the presence of HBsAg and anti-HBs. 

3. The decline in HBsAg prevalence after the initiation of universal HBV vaccination in Taiwan is not surprising; the results indicate the importance of vaccination in reducing the prevalence of chronic hepatitis B

4. The most interesting part of the paper is the changes in anti-HBs antibody positivity depending on the type of HBV vaccine used. Surprising is the decrease in anti-HBs positivity in the cohort of students vaccinated with recombinant HBV vaccine - this would merit a more detailed analysis.

5. In contrast, it is not surprising that increased ALT activity would be seen in obese, male and HBsAg positive individuals. This part of the paper seems unnecessary.

Round 2

Reviewer 1 Report

The modifications and additions made have sufficiently improved the quality of the article, therefore it can be recommended for publication in Vaccines.

Author Response

Thank you very much for your help in reviewing our manuscript.

Reviewer 2 Report

The authors have enhanced the manuscript.

Line 108-110, Please better clarify the method. I am not sure what authors mean to “clarify the impact of independent variables”. What was the outcome of the multivariate analysis? Was it the HBsAg status or the outcome is ALT, AST, etc. What were the covariates in the model or models?

Line 177-180, the information should in the method section. Were multivariable logistic conducted separately based on two outcomes variables (i.e., abnormal ALT and abnormal ALT of ASSLD criteria)?

Line 134-136, if I am reading it correctly, this is referred to Figure 1. The authors might want to specify the Figure when switching.

Line 242, based on the title of Table 3, the difference between the lab abnormal ALT and using ASSLD criteria is the cut-off level. Besides cut-off level, are there other difference? Is it expected to get similar results?

Line 286, As the author mentioned in line 69-71, the surveillance report for HBV vaccination program in Taiwan for every 5 years and focus in Taipei area. How were the results in the current study compared to the surveillance report?  Were the results consistent with the surveillance report?

Line 223-304, this is a very long paragraph with a lot of information. I suggest the authors to break it down to several paragraphs for clarity.

Author Response

Reviewer 2:

The authors have enhanced the manuscript.

Line 108-110, Please better clarify the method. I am not sure what authors mean to “clarify the impact of independent variables”. What was the outcome of the multivariate analysis? Was it the HBsAg status or the outcome is ALT, AST, etc. What were the covariates in the model or models?

Ans: Thanks for the comment. Univariate analyses were performed first in variables studied in descriptive statistics (age, gender, ALT, AST, total cholesterol, body height, body weight, and BMI) to identify the factors related to HBsAg status. A multivariate analysis was used then with the same variables as covariates in continuous form. BMI was the function of body height and body weight, and it would be the variable included in the model of multivariate analysis. We have addressed these points on lines 111-116

Line 177-180, the information should in the method section. Were multivariable logistic conducted separately based on two outcomes variables (i.e., abnormal ALT and abnormal ALT of ASSLD criteria)?

Ans: Thanks for the comment. Yes, two outcome variables were conducted separately. To clarify if it was clinically abnormal, we then compared the results of abnormal ALT of ASSLD criteria with the results of abnormal ALT. Two outcome variables with laboratory abnormal ALT cut-off level ( > 44 U/L) and abnormal ALT of ASSLD cut-off level (female > 25; male: > 35 U/L) were conducted separately. We have addressed these points on lines 121-123

Line 134-136, if I am reading it correctly, this is referred to Figure 1. The authors might want to specify the Figure when switching.

Ans: Thank you very much for the comment. When comparing trends among genders, for male students, the seropositivity of HBsAg was 11.54% for those who were born in 1985, and it decreased to less than 1% after 1991. The seropositivity of HBsAg in the female population decreased from 4% in the birth year of 1985 to less than 1% since the birth year 1991 (Figure 1). From the birth year 1985 to 2002, the trend of decreasing HBV immunity from 73.1% to 38.6% was noted in males. The same tendency for the decline was also noted in females from 76.0% to 37.9% from the birth year of 1985 to 2002. Besides, we also found transient increasing trends in 1987, 1990, 1991, 2000, and 2001 in the male group. In the female group, this tendency was noted in 1989, 1990, 1991, and 2000 (Figure 2). We have modified these points on lines 142-150.

Line 242, based on the title of Table 3, the difference between the lab abnormal ALT and using ASSLD criteria is the cut-off level. Besides cut-off level, are there other difference? Is it expected to get similar results?

Ans: Thanks for the comment. The laboratory abnormal ALT cut-off level ( > 44 U/L) was a general and non-specific warning to screen liver function in a health examination. For purposes of guiding the management of chronic hepatitis B (CHB), an upper limit of normal for ALT of 35 U/L for males and 25 U/L for females is recommended according to AASLD 2018 Hepatitis B Guidance. We have added on lines 118

Line 286, As the author mentioned in line 69-71, the surveillance report for HBV vaccination program in Taiwan for every 5 years and focus in Taipei area. How were the results in the current study compared to the surveillance report?  Were the results consistent with the surveillance report?
Ans: Yes. It is consistent. A decreasing trend of HBsAg seropositivity rate and a decreasing tendency of anti-HBs seropositivity rate in the postvaccination cohort. Furthermore, we minimized the age effect of immunity waning in the school-age population of the college.

Line 223-304, this is a very long paragraph with a lot of information. I suggest the authors to break it down to several paragraphs for clarity.

Ans: Thank you very much again. We have revised it according to your important comment.